# FabricFlowNet: Bimanual Cloth Manipulation with a Flow-based Policy

**Thomas Weng, Sujay Bajracharya, Yufei Wang, Khush Agrawal, and David Held**
Robotics Institute, Carnegie Mellon University, USA
{tweng, sbajrach, yufeiw2, khusha, dheld}@andrew.cmu.edu

**Abstract:** We address the problem of goal-directed cloth manipulation, a challenging task due to the deformability of cloth. Our insight is that optical flow, a technique normally used for motion estimation in video, can also provide an effective representation for corresponding cloth poses across observation and goal images. We introduce FabricFlowNet (FFN), a cloth manipulation policy that leverages flow as both an input and as an action representation to improve performance. FabricFlowNet also elegantly switches between dual-arm and single-arm actions based on the desired goal. We show that FabricFlowNet significantly outperforms state-of-the-art model-free and model-based cloth manipulation policies. We also present real-world experiments on a bimanual system, demonstrating effective sim-to-real transfer. Finally, we show that our method generalizes when trained on a single square cloth to other cloth shapes, such as T-shirts and rectangular cloths. Video and other supplementary materials are available at: https://sites.google.com/view/fabricflownet.

**Keywords:** deformable object manipulation, optical flow, bimanual manipulation

## 1 Introduction

Cloth manipulation has a wide range of applications in domestic and industrial settings. However, it has posed a challenge for robot manipulation: compared to rigid objects, fabrics have a higher-dimensional configuration space, can be partially observable due to self-occlusions in crumpled configurations, and do not transform rigidly when manipulated. Early approaches for cloth manipulation relied on scripted actions; these policies are typically slow and do not generalize to arbitrary cloth goal configurations [27, 13, 3].

Recently, learning-based approaches have been explored for cloth manipulation [18, 33, 42, 29, 32], including model-free reinforcement learning to obtain a policy [21, 39]. For a cloth manipulation policy to be general to many different objectives, it must receive a representation of the current folding objective. A standard approach for representing a goal-conditioned policy is to input an image of the current cloth configuration together with an image of the goal [21, 32].

We will show a number of downsides to such an approach when applied to cloth manipulation. First, the policy must learn to reason about the relationship between the current observation and the goal, while also reasoning about the action needed to obtain that goal. These are both difficult learning problems; requiring the network to reason about them jointly exacerbates the difficulty. Additionally, previous work has used reinforcement learning (RL) to try to learn such a policy [21, 39]; however, a reward function is a fairly weak supervisory signal, which makes it difficult to learn a complex cloth manipulation policy. Finally, while many desirable folding actions are more easily and accurately manipulated with bimanual actions, previous learning-based methods for goal-conditioned cloth manipulation have been restricted to single-arm policies.

In this paper, we introduce FabricFlowNet (FFN), a goal-conditioned policy for bimanual cloth manipulation that uses optical flow to improve policy performance (see Fig. 1). Optical flow has typically been used for video-related tasks such as object tracking and estimating camera motion. We demonstrate that flow can also be used in the context of policy learning for cloth manipulation; we use an optical flow-type network to estimate the relationship between the current observation and a sub-goal. We use flow in two ways: first, as an input representation to our policy; second, after estimating the pick points for a pick-and-place policy, we query the flow image to determine the place

5th Conference on Robot Learning (CoRL 2021), London, UK.

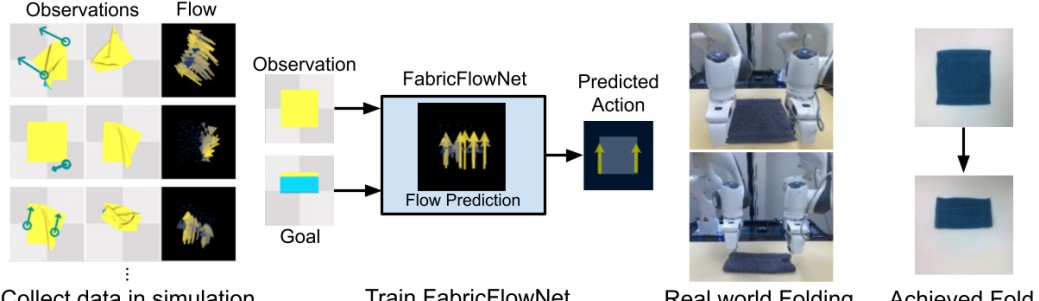

Figure 1: FabricFlowNet (FFN) overview. We collect a dataset of random actions and ground truth flow to train FFN. FFN learns to predict flow and uses it as both an input and action representation in a manipulation policy. FFN successfully performs single and dual-arm folding in the real world.

actions. Our method is learned entirely with supervised learning, leveraging ground truth particles from simulation. Our method learns purely from random actions without any expert demonstrations during training and without reinforcement learning.

Our learned policy can perform bimanual manipulation and switches easily between dual and single-arm actions, depending on what is most suitable for the desired goal. Our approach significantly outperforms our best efforts to extend recent single-arm cloth manipulation approaches to bimanual manipulation tasks [18, 21]. We present experiments on a dual-arm robot system and in simulation evaluating our method's cloth manipulation performance. FabricFlowNet outperforms state-of-the-art model-based and model-free baselines, and we provide extensive ablation experiments to demonstrate the importance of each component of our method to the achieved performance. Our method also generalizes with no additional training to other cloth shapes and colors. This paper contributes:

- A novel flow-based approach for learning goal-conditioned cloth manipulation policies that can perform dual-arm and single-arm actions
- A test suite for benchmarking goal-conditioned cloth folding algorithms encompassing and expanding on goals used in previous literature [18, 21, 15]; we perform extensive experiments using this test suite to evaluate FabricFlowNet (FFN), baselines [18, 21], and ablations, demonstrating that FFN outperforms previous approaches.
- Experiments to demonstrate that FFN generalizes to other cloth colors and shapes, even without training on such variations.

## 2   Related Work

**Bimanual Manipulation**. A large body of research exists on dual-arm, or bimanual, manipulation [35]. Dual-arm systems allow for more complex behaviors than single-arm systems at the cost of greater planning complexity [14, 34], leading to research on closed kinematic chain planning [36, 4], composable skill learning [40, 7], and rewarding synergistic behavior [8]. Prior work has also explored bimanual cloth manipulation [31], including establishing a diverse set of benchmark tasks [16]. Cloth manipulation is a highly underactuated task, and bimanual manipulation enables controlling multiple cloth points [5]. A common approach for cloth flattening is to lift a cloth with one arm and regrasp it with the other arm until it reaches the flattened configuration [23, 10, 27, 13, 3]. Previous work in this direction uses hard-coded policies [27, 13, 3], whereas we learn to achieve arbitrary folded configurations. Tanaka *et al.* [37] learn bimanual actions for goal-conditioned folding, using a voxel-based dynamics model to predict how actions will change the cloth state. However, optimizing this dynamics model can slow down inference time compared to our model-free approach. Dynamic bimanual manipulation has also been explored in simulation from ground-truth keypoints [20] and for unfolding cloth in the real world [17]; we perform real-world bimanual folding using depth image observations.

**Learning for Cloth Manipulation.** Prior works have proposed various hand-defined representations for cloth manipulation, such as parameterized shape models [28] or binary occupancy features [22]. Recent approaches use contrastive learning to learn pixel-wise latent embeddings for

cloth [15, 6]. Both contrastive learning [15] and goal-conditioned transporter networks [32] have been applied to imitate expert demonstrations. Our approach doesn't require expert actions, just sub-goal states provided at test-time to define the task. In contrast to these previous representations, our method uses a flow-based representation, which we found to significantly outperform previous methods for goal-based cloth manipulation.

Other approaches have applied policy learning techniques to single-arm cloth smoothing [39, 33]. In contrast, we learn a policy that performs either single and dual-arm cloth manipulation; further, our focus is on goal-conditioned cloth folding, rather than smoothing. For cloth manipulation, Lee *et al.* [21] learns a model-free value function, but is limited by its discrete action space, and further, they do not use a flow-based representation, which we show leads to large benefits. Prior methods for learning goal-conditioned policies have used self-supervised learning to learn an inverse dynamics model for rope [29, 30] but such approaches have not been demonstrated for cloth manipulation. Lippi *et al.* [25] plan cloth folding actions in latent space, but do not demonstrate generalization to unseen cloth shapes. Other papers use an online simulator [23], or learn a cloth dynamics model in latent space [42], pixel-space [18], or over a graph of keypoints [26]. Unlike these model-based methods, our method is model-free and does not require online simulation or time-expensive CEM planning, leading to much faster inference. Further, we compare our approach to state-of-the-art approaches for cloth manipulation [21, 18] and show significantly improved performance.

**Optical Flow for Policy Learning.** Optical flow is the task of estimating per-pixel correspondences between two images, typically for video-related tasks such as object tracking and motion estimation. State-of-the-art approaches use convolutional neural networks (CNN) to estimate flow [12, 19, 38]. Optical flow between successive observations has previously been used as an input representation to capture object motion for peg insertion [11] or dynamic tasks [1]. Within the domain of cloth manipulation, Yamazaki *et al.* [41] similarly use optical flow on successive observations to identify failed actions. We use flow not to represent motion between successive images, but to correspond the cloth pose between observation and goal images, and to determine the placing action for folding. Argus *et al.* [2] use flow in a visual servoing task to compute residual transformations between images from a demonstration trajectory and observed images. In contrast, we learn a policy with flow to determine what cloth folding actions to take, not how to servo to a desired pose.

## 3 Learning a Goal-Conditioned Policy for Bimanual Cloth Manipulation

### 3.1 Problem Definition

Our objective is to enable a robot to perform cloth folding manipulation tasks. Let each task be defined by a sequence of sub-goal observations $\mathcal{G} : \{x_1^g, x_2^g, \ldots, x_N^g\}$, each of which can be achieved by a single (possibly bimanual) pick-and-place action from the previous sub-goal. We require sub-goals, rather than a single goal, because a folded cloth can be highly self-occluded such that a single goal observation fails to describe the full goal state. Defining a task using a sequence of sub-goals is found in other recent work [30]. Similar to prior work [30, 29], even if the sub-goals are obtained from an expert demonstration, we nonetheless do not assume access to the expert actions; this is a realistic assumption if the sub-goals are obtained from visual observations of a human demonstrator.

We assume that the agent does not have access to the sub-goal sequence $\mathcal{G}$ during training that it must execute during inference. Thus, the agent must learn a general goal-conditioned policy $a_t = \pi(x_t, \mathcal{G})$, where $x_t$ is the current observation of the cloth and $a_t \in \mathcal{A}$ is the action selected by the policy. In our approach, we input each sub-goal $x_i^g$ sequentially to our policy: $a_t = \pi(x_t, x_i^g)$. A goal recognizer [30] can also be used to decide which sub-goal observation to input at each timestep. For convenience, we will interchangeably refer to $x_i^g$ as a goal or sub-goal.

### 3.2 Overview

A common approach for a goal-conditioned policy is to input the current observation $x_t$ and the goal observation $x_i^g$ directly into to a neural network representation of a policy [30, 21] or a Q-function [39, 21]. However, the network must reason simultaneously about the relationship between the observation and the goal, as well as the correct action to achieve that goal. Our first insight is that we can improve performance by separating these two components: we will learn to reason about the relationship between the observation and the goal, and separately use this relationship to reason

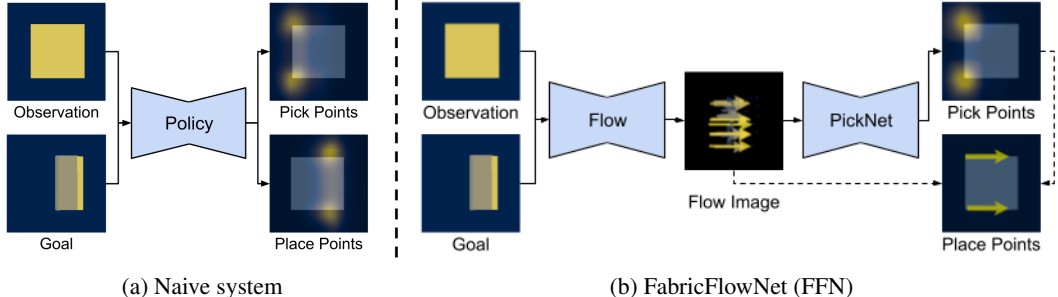

(a) Naive system          (b) FabricFlowNet (FFN)

Figure 2: (a) A naive approach to goal-conditioned policy learning is to input observation and goal images directly to the policy and predict the action. (b) FabricFlowNet separates representation learning from policy learning; it first estimates the correspondence between the observation and goal as a flow image. The flow is then used as the input to PickNet for pick point prediction, and as a way to compute place points without requiring additional learning.

over actions. Specifically, we represent this relationship using a "flow image" $f$, which indicates the correspondence between the current observation $x_t$ and sub-goal $x_i^g$. Thus we propose using the flow image $f$ as an improved input representation of the policy, rather than directly inputting the observation $x_t$ and goal observation $x_i^g$.

Our second insight is that we can also use flow in the output representation of the policy. We use a pick and place action space; prior methods that learn pick and place policies for deformable object manipulation predict place points using the policy network, either explicitly [33, 32, 42, 29, 30, 39] or implicitly by transforming the inputs to a Q-function [21]. Instead, we simplify the problem by leveraging flow: our policy network only learns to predict the *pick* points. For the place point, we query the flow image $f$ for the flow vector starting at the predicted pick location, and use the endpoint of that vector as the place point.

We demonstrate that using flow in the two ways described above for our policy achieves significantly improved performance compared to prior work. Furthermore, our approach extends naturally to dual-arm manipulation, allowing us to easily transition between single and dual-arm actions.

A schematic overview of our system can be found in Fig. 2b. We first compute the flow $f$ between the current observation $x_t$ and goal $x_i^g$. Next, we input the flow $f$ to a policy network (PickNet), which outputs pick points $p_i$. We then query the flow image $f(p_i)$ to determine the place points for each robot arm. Further details of our approach are described below.

### 3.3 Estimating Flow between Observation and Goal Images

We learn flow to use it as an input representation to our pick prediction network, and as an action representation for computing place points. Given an observed depth image $x_t$ and desired goal depth image $x_i^g$, we estimate the flow $f = (f^1, f^2)$, mapping each pixel $(u, v)$ in $x_t$ to its corresponding coordinates $(u', v') = (u + f^1(u), v + f^2(v))$ in $x_i^g$. This task formulation differs from standard optical flow tasks as the input image pairs $(x_t, x_i^g)$ are not consecutive images from video frames.

To capture the complex correspondences between $x_t$ and $x_i^g$, we train a convolutional neural network to estimate the flow image $f$ (see Appendix for details). The training loss we use to supervise the network is endpoint error (EPE), the standard error for optical flow estimation. EPE is the Euclidean distance between the predicted flow vectors $f$ and the ground truth $f^*$, averaged over all pixels: $\mathcal{L}_{\text{EPE}} = \frac{1}{N} \sum_{i=1}^{N} \|f^* - f\|_2$. We use a cloth simulation to collect training examples with ground truth flow. The simulator provides the ground-truth correspondence between the particles of the cloth in different poses. The simulation cloth particles are not as dense as the depth image pixels; as a result, we only have ground-truth flow supervision for a sparse subset of the pixels that align with the cloth particles. Thus, we mask the loss to only supervise the flow for the pixels that align with the location of the cloth particles. We train the flow network using data collected from random actions. See Sec. 3.6 for more details on the simulator, data collection, and network training.

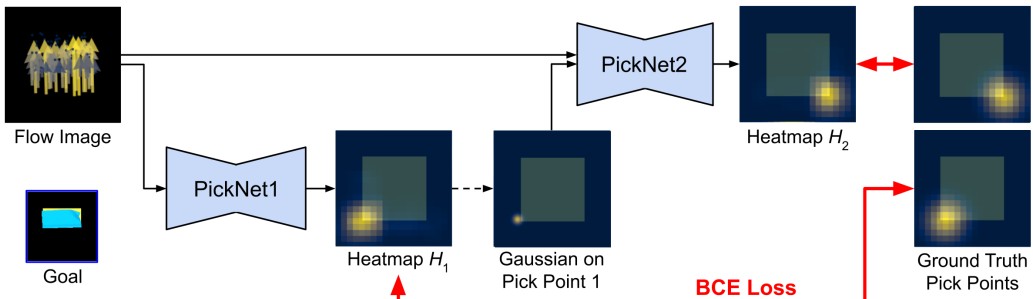

Figure 3: PickNet architecture. We utilize a two-network architecture for bimanual manipulation, where the second pick point is conditioned on the prediction of the first pick point.

## 3.4 Learning to Predict Pick Points

Our bimanual action space $\mathcal{A}$ consists of actions $a = (p_1, p_2, q_1, q_2)$, where $p$ and $q$ are the pick and place points respectively, paired according to the subscripts. We train a neural network called PickNet to estimate the pick points $p_1, p_2$. Crucially, the input to PickNet is a flow image $f$, estimated between the current depth image $x_t$ and the desired goal depth image $x_i^g$, as described in the previous section. The flow image indicates, for each pixel $(u, v)$ in the current observation, the location $f(u, v)$ that the pixel has moved to in the goal observation. Our flow network (Sec. 3.3 above) reasons about the observation-goal relationship, so that the policy network (PickNet) only needs to reason about the action, specifically the two pick points $(p_1, p_2)$; computing the place points is described in Sec. 3.5.

For dual-arm actions, the pick points must be estimated conditionally, as the location of pick point $p_1$ on the cloth influences the optimal location of pick point $p_2$, and vice versa. To decouple this conditional estimation problem, we propose a two-network architecture, PickNet1 and PickNet2, to estimate the pick points (see Fig. 3). This architecture was inspired by Wu *et al.* [39], which used two networks for pick-conditioned placing; we instead use two networks to condition dual-arm picking. PickNet1 is a fully convolutional network that receives flow image $f$ as input and outputs a single heatmap $H_1$ estimating the optimal pick points for arm 1. We compute the first pick point as $p_1 = \arg\max_p H_1(p)$. The second network, PickNet2, predicts the second arm's pick point $p_2$ conditioned on $p_1$; PickNet2 takes as input both the flow image $f$ and an additional image with a 2D Gaussian centered on $p_1$, and is otherwise identical to PickNet1. PickNet2 outputs heatmap $H_2$, from which we compute the second pick point: $p_2 = \arg\max_p H_2(p)$. The two-network architecture decouples the conditionally dependent pick point predictions and does not require us to resort to heuristics to extract two pick points from a single heatmap. We refer to PickNet1 and PickNet2 together as "PickNet."

To train PickNet, we collect a dataset of random actions (see Sec. 3.6 for details) and record the current observation $x_t$, the bimanual action $a = (p_1, p_2, q_1, q_2)$, and the next observation $x_{t+1}$. We also estimate the flow $f$ from $x_t$ to $x_{t+1}$, as explained in Sec. 3.3. We create ground truth pick heatmaps $H_i^*$ for arm $i$ using the recorded random action $a$, by placing a 2D Gaussian $\mathcal{N}(p_i, \sigma)$ on each ground truth pick location $p_i$. We then supervise PickNet using the binary cross-entropy (BCE) loss between predicted heatmaps $H_1, H_2$ and ground truth heatmaps $H_1^*, H_2^*$. However, it might be unclear to the network which pick point should be output by PickNet1 and which should be output by PickNet2. We compute the loss for both possible correspondences and use the minimum:

$$l_{\text{BCE}}(H_i, H_j, H_i^*, H_j^*) = \text{BCE}(H_i, H_i^*) + \text{BCE}(H_j, H_j^*)$$
$$\mathcal{L}_{\text{Pick}} = \min[l_{\text{BCE}}(H_1, H_2, H_1^*, H_2^*), l_{\text{BCE}}(H_2, H_1, H_1^*, H_2^*)]$$

$$(1)$$

At inference time, PickNet outputs the pick points $p_1, p_2$, computed from the argmax of $H_1, H_2$ respectively, as described above.

## 3.5 Estimating the Place Points from Flow

After estimating the pick points $p_1, p_2$ from flow, the remaining step to predict a bimanual pick and place action $a = (p_1, p_2, q_1, q_2)$ is to estimate the place points $q_1, q_2$. A straightforward approach would be to train the network to predict place points $q_1, q_2$, similar to the pick points $p_1, p_2$ as

described above (see Fig. 2a). Instead, our approach uses the flow image to find the place points, so that the place points do not have to be learned separately.

Our approach makes the assumption that, to achieve a desired subgoal configuration, the point picked on the cloth should be moved to its corresponding position in the goal image (which is estimated by the flow). This is a simplifying assumption, since it is possible that the picked point will shift slightly after it is released by the gripper; our method does not take into account such small movements. Using this assumption, to compute the place points $q_1, q_2$, we query the flow $f$ at each pick point $p_1, p_2$ to estimate the delta between the pick point location in the observation image and the corresponding location of the pick points in the goal image. We use these predicted correspondences as the place points: $q_i = f(p_i) + p_i$, for each arm $i$.

Action predictions estimated by our approach can produce nearly overlapping pick and place points, indicating that arm 1 and arm 2 should perform identical actions. We observe this behavior from PickNet when the goal is best achieved with a single-arm action, rather than a bimanual one. On a real robot, grippers are likely to collide if grasping points that are too close. Therefore, to switch between executing a single-arm or bimanual action, we compute the L2 pixel distance between pick points $d_{\text{pick}} = \|p_1 - p_2\|_2$ and place points $d_{\text{place}} = \|q_1 - q_2\|_2$. We use a single-arm action when either distance is smaller than a threshold $\alpha$, which we set to 30 for all experiments.

### 3.6 Implementation Details

We use SoftGym [24], an environment for cloth manipulation built on the particle-based simulator Nvidia Flex, to collect training datasets. The simulator models cloth as particles connected by springs. We use pickers that simulate a grasping action by binding to the nearest cloth particle within a threshold to execute pick and place actions in SoftGym. We collect data by taking random actions, biased towards grasping corners of the cloth. We demonstrate that we are able to train our method in SoftGym and then transfer the policy to the real world. Details on the data collection, as well as the network architecture and training details, can be found in Appendix Sec. A.1.

## 4 Experiments

### 4.1 Simulation Experiments

**Experiment Setup.** We evaluate FabricFlowNet (FFN) and compare to state-of-the-art baselines in the SoftGym [24] simulator; real-world evaluations are below in Sec. 4.2. Our experiments focus on folding tasks, and we assume that a cloth smoothing method (*e.g.,* [17, 33]) is used to flatten the cloth before folding is executed. Our error metric is the average particle position error between the achieved and goal cloth configuration. We evaluate on two sets of goals: 40 *one-step* goals that can be achieved with a single fold action, and 6 *multi-step* goals that require multiple folding actions. The multi-step goals each consist of a sequence of sub-goal images, with the next sub-goal presented after each action. This protocol follows from our problem formulation in Sec. 3.1, and is similar to the protocol in Nair *et al.* [29]. Our goals include test goals from Ganapathi *et al.* [15] and Lee *et al.* [21] that are achievable with one arm, as well as additional goals more suitable for two-arm actions (see Appendix Fig. S2 for the full set of goals).

We compare our method to Fabric-VSF [18], which learns a visual dynamics model and uses CEM to plan using the model. We only use Fabric-VSF with RGB-D input, as depth-only input performs poorly for folding tasks [18]. FabricFlowNet only uses depth and does not rely on RGB, which enables our method to transfer easily to the real world without extensive domain randomization. We also compare to Lee *et al.* [21], a model-free approach. We extend the the original single-arm method to a dual-arm variant and compare against both. For both our method and the baselines, we only allow each method to perform one pick-and-place action for each subgoal (e.g. one pick and place action for each single-step goals). Additional baseline details can be found in the Appendix.

### 4.1.1 Simulation Results

Table 1 contains our simulation results for all methods. We report average particle distance error (in mm) for one-step goals only, multi-step goals only, and over both one-step and multi-step goals. Our results show that FFN achieves the lowest error over all goals and has the fastest inference time.

Table 1: Mean Particle Distance Error (mm) and Inference Time (sec) on Cloth Folding Goals

| Method | One Step (n=40) | Multi Step (n=6) | All (n=46) | Inference Time |
|---|---|---|---|---|
| Lee *et al.*, 1-Arm [21] | $16.18 \pm 08.38$ | $26.20 \pm 16.31$ | $17.49 \pm 10.10$ | $\sim 0.04$ |
| Lee *et al.*, 2-Arm | $36.62 \pm 14.51$ | $47.71 \pm 21.95$ | $38.07 \pm 15.82$ | $\sim 0.04$ |
| Fabric-VSF [18] | $6.31 \pm 06.55$ | $\mathbf{21.33 \pm 11.20}$ | $8.27 \pm 08.90$ | $\sim 420$ |
| FabricFlowNet (Ours) | $\mathbf{4.46 \pm 02.62}$ | $25.04 \pm 22.88$ | $\mathbf{7.14 \pm 11.06}$ | $\mathbf{\sim 0.007}$ |

We also investigate whether using flow as a goal recognizer improves performance. When an observation closely matches the goal, the flow for all points is close to zero. We leverage this fact by evaluating FFN with "iterative refinement": we allow the policy to take multiple actions per subgoal to try to further minimize the flow between the observation and subgoal. When the average flow between observation and current subgoal reaches a minimum threshold, the policy moves forward to the next subgoal. FFN with iterative refinement achieves a mean error of 6.62 over all goals vs. 7.14 without refinement. Additional details on iterative refinement can be found in the Appendix, along with additional results from baseline variants, crumpled initial configurations, and end-to-end training.

### 4.1.2 Ablations

We run series of ablations to evaluate the importance of the components of our system; results averaged over all 46 goals are in Table 2. Additional details and results are in Appendix Sec. D. Our ablations are designed to answer the following questions:

**What is the benefit of using flow as input?** We modify PickNet to receive depth images of the observation and goal as input to the network ("NoFlowIn"), as is commonly done in previous work on goal-conditioned RL [21, 32]. In this ablation, the PickNet needs to reason about both the relationship between the observation and the goal, as well as the action. In contrast, our method uses the flow network to compare the observation and goal; the picknet separately reasons about the action.
**What is the benefit of using flow to choose the place point?** In this ablation, we train a network to predict the place points directly ("NoFlowPlace"). This is in contrast to our approach where we use the flow field, evaluated at the pick point $f(p_i)$, to compute the place point $q_i$ for arm $i$. Our approach leads to a 32.4% improvement, showing the benefit to using flow as an action representation.
**What is the performance with no flow?** We combine the above two ablations and remove flow entirely, ("NoFlow"; ours has 60.4% improvement). The above ablations all indicate the strong benefit of using flow as both an input and action representation for cloth manipulation.
**What is the benefit of biasing the data collection to grasp corners?** Our method uses prior knowledge about cloth folding tasks to bias the training data and pick at corners of the cloth. In this ablation, we choose pick points randomly ("NoCornerBias", ours has 35.5% better performance).
**What is the performance with a simpler architecture?** We also compare our architecture for PickNet (Sec. 3.4) to a simpler architecture that takes as input the flow image $I_f$ and outputs a two heatmaps, one for each pick point ("NoSplitPickNet"; ours has 2.1% better performance).
**Does the loss formulation in Eq. 1 improve performance?** We compare our method to an ablation where the first ground-truth heatmap is used to supervise PickNet1 and similarly for the second, i.e. $\mathcal{L}_{\text{Pick}} = l_{\text{BCE}}(H_1, H_2, H_1^*, H_2^*)$. ("NoMinLoss"; ours has similar performance).

Table 2: Mean Particle Distance Error (mm) for Ablations over All Goals (n=46)

| NoFlowIn | NoFlowPlace | NoFlow | NoCornerBias | NoSplitPickNet | NoMinLoss | FFN (Ours) |
|---|---|---|---|---|---|---|
| 9.37 | 10.56 | 18.02 | 11.07 | 7.29 | 7.15 | **7.14** |

## 4.2 Real World Experiments

We evaluate FabricFlowNet in the real world and demonstrate that our approach successfully manipulates cloth on a real robot system.

**Experiment Setup.** Our robot system consists of two 7-DOF Franka Emika Panda arms and a single wrist-mounted Intel RealSense D435 sensor (See Fig. 1). We plan pick and place trajectories using MoveIt! [9]. We evaluate on a 30x30 cm towel, using 6 single-step and 5 multi-step goals (see Fig. 4) that form a representative subset of our simulation test goals.

To transfer from simulation to the real world, we align the depth between real and simulated images by subtracting the difference between the average depth of the real support surface (i.e. the table) and the simulated surface. We mask the cloth by color-thresholding the background; see Appendix for details. We found that these simple techniques were sufficient to transfer the method trained entirely in simulation to the real world, because we use only depth images as input. Simulated depth images match reasonably well to real depth images, unlike RGB images.

### 4.2.1 Real World Results

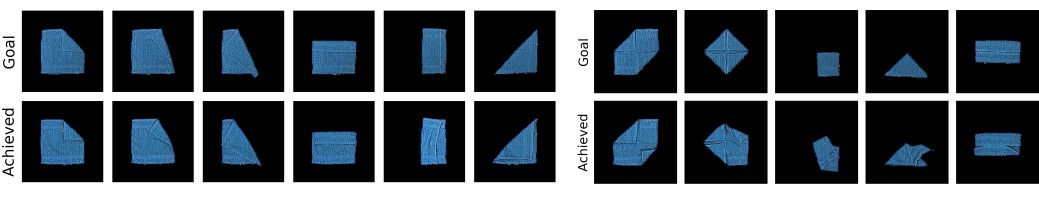

(a) One-step Square Cloth  (b) Multi-step Square Cloth

Figure 4: Qualitative results for FFN on real world experiments. FFN only takes depth images as input, allowing it to easily transfer to cloth of different colors.

Fig. 4 provides qualitative real world results, showing that we successfully achieve many of the goals. Our website (link in abstract) contains videos of these trials.

We compare FabricFlowNet to the NoFlow ablation from Sec. 4.1.2. Both methods used the same sim-to-real techniques described in the previous section. While we do not have access to the true cloth position error in the real world, Intersection-over-Union (IoU) on the achieved cloth masks serves as a reasonable proxy metric [21]. FFN achieves 0.80 mean IoU over 3 trials for the square cloth, compared to 0.53 for NoFlow. See the Appendix for additional details.

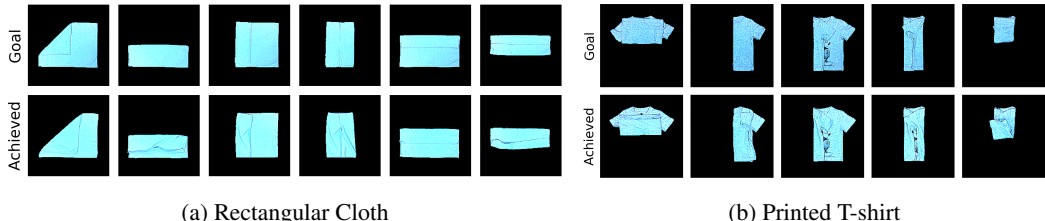

(a) Rectangular Cloth  (b) Printed T-shirt

Figure 5: Generalization to new cloth shapes for FFN trained only on a square cloth in simulation. FFN achieves single and multi-step goals for rectangular fabric and a printed T-shirt.

**Generalization.** In addition to evaluating the folding policy on square cloth for various goal configurations, we also test the generalization of our method to other shapes of cloth. We evaluate the performance of FFN trained only on a square cloth on folding goals for a rectangular cloth as well as a T-shirt. These fabrics are also thinner than the square blue towel used in the real world experiments above. Fig. 5 shows that FFN trained on a square yellow cloth in simulation is able to generalize to other cloth shapes, textures, and colors (FFN only receives depth images as input). See Appendix for additional details.

## 5  Conclusion

In this work we present FabricFlowNet, a method which utilizes flow to learn goal-conditioned fabric folding. We leverage flow to represent the correspondence between observations and goals, and as an action representation. The method is trained entirely using random data in simulation. Our results show that separating the correspondence learning and the policy learning can improve performance on an extensive suite of single- and dual-arm folding goals in simulated and real environments. Our experiments also demonstrate generalization to different fabric shapes, textures, and colors. Future work on flow-based fabric manipulation could incorporate actions beyond pick and place, such as parameterized trajectories or dynamic actions.

**Acknowledgments**

This work was supported by the National Science Foundation (NSF) Smart and Autonomous Systems Program (IIS-1849154), LG Electronics, and a NSF Graduate Research Fellowship (DGE-1745016).

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
