# OpenReview forum: "FabricFlowNet: Bimanual Cloth Manipulation with a Flow-based Policy"
_robot-learning.org/CoRL/2021/Conference — CoRL2021 Poster_

### Official Review · Reviewer_sL8a · 2021-07-20

**Originality:** Good
**Technical Quality:** Good
**Clarity Of Presentation:** Very Good
**Impact:** 3

**Recommendation:**

Weak Accept: I recommend accepting the paper, but will not argue for my recommendation if the majority of other reviewers have a different opinion.

**Summary:**

The authors proposing using dense point-to-point cloth correspondence (flow) as a representation for cloth manipulation. The authors propose training a flow model in simulation which produces point correspondences between observations (initial configurations) and goal states of the cloth. From this representation they then predict pick points and use the predicted flow to implicitly determine placement locations, resulting in a final pick-and-place style policy. They show convincing results in a bimanual tabletop environment with two Panda arms.

Update (Post Author Response):
I appreciate the author's response to my questions; fundamentally my opinion of this paper still stands: It proposes an interesting and practical representation for cloth manipulation and they present fairly compelling evidence for its efficacy in the regime that it was designed for (fairly nicely behaved scenes and initial conditions). I think this paper is a useful addition to the field and advocate for its acceptance.

**Issues:**

I have no major issues that I need addressed. In a vacuum (i.e. not having discussed with other reviewers) would advocate for acceptance in the paper's current state.

**Reviewer Expertise:**

Very good: Comprehensive knowledge of the area

**Strengths And Weaknesses:**

The main strengths of this work lie in its simplicity and effectiveness. The authors show that simple point-to-point correspondence, trained in sim, generalizes well to the real world and results in a usable system (even ~2cm of placement error in the multi-step version is pretty impressive compared to prior work). The authors also nicely ablate the differences between the flow-based input and the flow-based output parameterization. Interestingly (and surprisingly for me), most of the benefit came from the output parameterization using flow. I also found the paper to be well written and clearly presented, without unnecessary pomp and circumstance to try to dress up the work (at the cost of clarity and understandability). This is refreshing and great to see!

The biggest weakness of this paper was the lack of dynamics incorporation. Generally, these sorts of pick and place policies are inefficient compared to policies that incorporate dynamic effects (such as grasping and shaking a cloth to flatten it before the next fold). This isn't a fatal flaw, but it would strengthen the paper to briefly discuss this limitation and how a flow-based representation might fit into a dynamically aware paradigm in the future.

Another question is how well the long-term temporal correspondence works when dealing with highly crumpled initial conditions. By relying so heavily on good long-horizon flow/correspondence predictions, it seems that this method might struggle with such cases (e.g. a crumpled ball of a shirt). It would also strengthen the paper to evaluate such examples (even if the answer is "this representation works less well in those cases").

**Summary Of Recommendation:**

This paper is relatively novel (correspondence-based representations and flow have been around for quite a while, as the authors themselves note, but their application to cloth folding in this way is novel). In the relatively controlled domain the authors explore, this approach appears effective and the work is well-written and includes useful, and in some cases surprising (see my reaction to the ablation section) information for other practitioners.

---

> ### Author Response · Authors · 2021-08-29
> **Response to Reviewer sL8a**
>
> Thank you for your insightful comments and suggestions. We address each point in detail below.
>
> __Q1__:
> > The biggest weakness of this paper was the lack of dynamics incorporation….it would strengthen the paper to briefly discuss this limitation and how a flow-based representation might fit into a dynamically aware paradigm in the future.
>
> Dynamic actions may be more useful for unfolding behaviors (as explored by Ha and Song [17]) than for folding, where static actions were suitable for the large set of goals we evaluated on. We are definitely interested in exploring the use of more complex action spaces with flow-based cloth manipulation. As per your suggestion, we have added exploring dynamic actions to the conclusion as an area of future work.
>
> __Q2__:
> > Another question is how well the long-term temporal correspondence works when dealing with highly crumpled initial conditions. By relying so heavily on good long-horizon flow/correspondence predictions, it seems that this method might struggle with such cases (e.g. a crumpled ball of a shirt). It would also strengthen the paper to evaluate such examples (even if the answer is "this representation works less well in those cases").
>
> Since our method uses flow to find correspondences between the observation and goal, it is true that starting with observations of highly crumpled cloth will perform significantly worse than starting with an already flattened cloth. Our experiments focused on folding tasks, and we assume that a previous method was used to flatten the cloth before our method is executed; a method such as Ha and Song [17] or Seita et al. [33] ([34] in the latest revision) could be used for this purpose. We have added a sentence to Section 4.1 to clarify this point. An area of potential future work is to connect these methods together (smoothing + folding) together into a single system.
>
> To evaluate the robustness of our method to imperfect smoothing, we performed additional experiments to evaluate the performance of our method from slightly crumpled initial configurations. These experiments can be found in Supplementary Section G. We create three slightly crumpled initial configurations (Supplementary Figure S5) and evaluate FFN on the full set of goals (Supplementary Figure S1a and S1e) when starting each trial from these crumpled configurations. We find that FFN performs slightly worse on particle distance error metrics (Supplementary Table S10), but qualitatively still achieves pretty good results (Supplementary Figure S6). Please refer to Supplementary Section G for additional details.

---

### Official Review · Reviewer_RKFg · 2021-07-24

**Originality:** Very Good
**Technical Quality:** Very Good
**Clarity Of Presentation:** Very Good
**Impact:** 3

**Recommendation:**

Weak Accept: I recommend accepting the paper, but will not argue for my recommendation if the majority of other reviewers have a different opinion.

**Summary:**

The paper contributes a novel and efficient framework for bimanual cloth manipulation. Specifically, the proposed method leverages the concept of optical flow to represent both the cloth pose and actions. Without relying of expert demonstrations or explicit reward signals, the proposed approach is shown to be capable of learning cloth manipulation policies that generalize to new cloth shapes without additional training.

**Issues:**

1. Please considering removing the last bullet from the list of contributions as evaluation of a proposed framework can not considered as a contribution. Having said that, it might be appropriate to list the ability of the proposed framework to general to other cloth colors and shapes as a contribution.
2. How robust is the proposed approach expected to be variations in the background?
3. It seems difficult to know how many sub-goals are necessary for a given task/goal. Can exiting methods (e.g., Ref. [33]) address this challenge?
4. For the ablation experiments, please include the error associated with the full method in Table 2 (currently, only relative improvements are reported in text and the actual error is only included in the supplementary material).
5. I could not quite understand the motivation for not including key information in Table S2 from the appendix in the manuscript (e.g., many variants of the Fabric-VSF baseline from this table do better than what is reported in Table I for Fabric-VSF). Why are the best performing variants of the baseline not included in the manuscript?
6. Unlike the proposed approach and Fabric-VSF, why was Lee et al.'s ability to generalize to new cloth shapes not evaluated?

**Reviewer Expertise:**

Good: General knowledge of the area

**Strengths And Weaknesses:**

**Strengths**
+ The paper addresses a timely and well-motivated problem.
+ The use of optical flow for cloth manipulation seems to be a novel and ingenious application of well understood concept from a different filed (computer vision / estimation).
+ The experimental design includes multiple tasks (including those that require multiple steps), ablation experiments, a real-robot platform, and comparisons against SOTA approaches to cloth manipulation. Further, the results clearly demonstrate the utility and relative benefits of the proposed approach over existing approaches.
+ The method neither relies on expert demonstrations nor explicit reward signal. It simply leverages random explorations to collect data for supervised learning of flow prediction
+ The method autonomously switches between single and dual arm behavior depending on the distance between pick (or place) points.

**Weaknesses**

- The related work section on "Policy Learning for Cloth Manipulation" merely lists model-based and model-free methods for cloth manipulation, and mentioned that experimental comparisons are carried out. A qualitative discussion on the similarities and differences (much like how it is done in other subsections of related work) would be a lot more useful here.
- The approach assumes that the appropriate number of sub goals required for any given goal is known and each of those sub-goals provided to the robot.
- The approach relies on a simulator for ground-truth. While this reliance is not a weakness in itself, the paper needs to make this more important. For instance, this reliance must be disclosed in the introduction.
- The paper does not cite or utilize an important related work (Garcia-Camacho et al., RAL, 2020) that contributes a comprehensive collection of evaluation metrics and benchmark scenarios for bimanual cloth folding that would have made the evaluation much stronger. Further, this prior somewhat weakness the second contribution of the paper relating to the test suite for benchmarking

**References**

- Garcia-Camacho, Irene, et al. "Benchmarking bimanual cloth manipulation." IEEE Robotics and Automation Letters 5.2 (2020): 1111-1118.

**Summary Of Recommendation:**

The paper proposes a novel and effective approach to learning bimanual cloth manipulation policies. Barring some relatively minor weaknesses in the experimental evaluation, the proposed method appears to be much more efficient and somewhat more effective than SOTA approaches.

---

> ### Author Response · Authors · 2021-08-29
> **Response to Reviewer RKFg (1/2)**
>
> Thank you for your insightful comments and suggestions. We address each point in detail below.
>
> __Q1__:
> > The related work section on "Policy Learning for Cloth Manipulation" merely lists model-based and model-free methods for cloth manipulation, and mentioned that experimental comparisons are carried out. A qualitative discussion on the similarities and differences (much like how it is done in other subsections of related work) would be a lot more useful here.
>
> We have revised the related work section to discuss similarities and differences with prior work.
>
> __Q2__:
> > The approach assumes that the appropriate number of sub goals required for any given goal is known and each of those sub-goals provided to the robot...It seems difficult to know how many sub-goals are necessary for a given task/goal. Can exiting methods (e.g., Ref. [33]) address this challenge?
>
> To clarify, our method requires sub-goals at test time, not training time; we train only on random actions and require no expert demonstrations for training. Subgoals are required at test time in order to fully specify the task. Many cloth folding goals have final goal configurations in which large portions of the cloth are self-occluded. Therefore, subgoals are required to ensure the task is completed correctly and that the cloth is correctly folded.
>
> While obtaining subgoals may seem difficult, they can be extracted from test-time human demonstrations which specify the folding task. Some form of human interaction is required to specify a folding task, so this is not a large additional burden in terms of human demonstration cost. We only require the observation of the intermediate goals from the human demonstration, and we do not need to infer the human actions. Since extracting subgoals from test-time demonstrations is somewhat orthogonal to the main contribution of our paper on using flow for manipulation, we provide the subgoal images directly.
>
> Seita et al. [33] (reference [34] in the updated revision) performs cloth smoothing, which does not require goal-conditioning, so this method is unlikely to work directly on goal-conditioned cloth folding without significant modifications. The oracle policy used to train the network pulls misaligned corners to align them, but otherwise does not use explicit subgoal reasoning such as taking actions to expose occluded corners. Lee et al. demonstrated cloth folding without subgoals by relying on the learned Q-value heatmap to select actions that would lead toward a single final goal. We provide additional experiments in Table S6 in Supplementary Section C.2 demonstrating that Lee et al. with subgoals achieves better performance than without using subgoals (17.49 vs. 19.71 mean particle distance). Furthermore, even when both methods use subgoals, Lee et al. does not perform as well as FFN on our test goal suite (17.49 vs 7.14 mean particle distance).
>
> Nonetheless, we recognize that, in certain cases, it may be too burdensome for the human demonstrator to provide subgoals; we leave handling such cases to future work; one possible direction is to combine our method with latent space roadmaps [26].
>
> __Q3__:
> > The approach relies on a simulator for ground-truth. While this reliance is not a weakness in itself, the paper needs to make this more important. For instance, this reliance must be disclosed in the introduction.
>
> In the original submission, Section 3.3 in the main paper describes using the ground truth cloth particles from the simulator to supervise our flow network. In the latest paper revision, we have also stated this in the introduction.
>
> __Q4__:
> > The paper does not cite or utilize an important related work (Garcia-Camacho et al., RAL, 2020) that contributes a comprehensive collection of evaluation metrics and benchmark scenarios for bimanual cloth folding that would have made the evaluation much stronger. Further, this prior somewhat weakness the second contribution of the paper relating to the test suite for benchmarking
>
> We do agree that Garcia-Camacho is an important missing citation in our related work. In the revised paper we have cited Garcia-Camacho in the “Bimanual Manipulation” related work heading.
>
> In terms of our contribution as a test suite for benchmarking, we are specifically focused on goals related to the cloth folding task. Our suite of goals is more extensive than that used in Fabric-VSF [18] and Lee et al [21] for cloth folding and we evaluate these baselines as well as our method and ablations on this extensive goal suite.
>
> __Q5__:
> > Please considering removing the last bullet from the list of contributions as evaluation of a proposed framework can not considered as a contribution. Having said that, it might be appropriate to list the ability of the proposed framework to general to other cloth colors and shapes as a contribution.
>
> We have edited the 3rd bullet as per your suggestion.

---

> > ### Author Response · Authors · 2021-08-29
> > **Response to Reviewer RKFg (2/2)**
> >
> > __Q6__:
> > > How robust is the proposed approach expected to be variations in the background?
> >
> > We obtain a background mask of the table using color-based HSV thresholding, which we can determine before the cloth is placed on the table; we then use the inverse of this background mask to obtain a mask of the cloth. Note that while we use background color of the table for cloth masking, the network itself only takes depth input, allowing the network to be robust to colors and patterns on the cloth itself. We have clarified this masking procedure by adding a paragraph to Supplementary Section A.3. A more sophisticated masking method could be more robust to background variation than the simple color-based thresholding used, but cloth masking or segmentation is orthogonal to our primary contribution on flow for fabric manipulation.
> >
> > __Q7__:
> > > For the ablation experiments, please include the error associated with the full method in Table 2 (currently, only relative improvements are reported in text and the actual error is only included in the supplementary material).
> >
> > We have included the error associated with the full method (FFN) in Table 2, and added a sentence to the beginning of Section 4.1.2 to clarify that the ablations are evaluated on the same set of goals as in Table 1.
> >
> > __Q8__:
> > > I could not quite understand the motivation for not including key information in Table S2 from the appendix in the manuscript (e.g., many variants of the Fabric-VSF baseline from this table do better than what is reported in Table I for Fabric-VSF). Why are the best performing variants of the baseline not included in the manuscript?
> >
> > We had some late-breaking results from variations of the baselines that came in after the main paper deadline, so we included those results in the supplement. In the latest paper revision, we have updated the results in Table 1 in the main paper to reflect the best variants of each method. Additional lesser-performing Fabric-VSF variants are still provided in the Supplementary Table S2.
> >
> > __Q9__:
> > > Unlike the proposed approach and Fabric-VSF, why was Lee et al.'s ability to generalize to new cloth shapes not evaluated?
> >
> > In the latest paper revision, we have evaluated the generalization ability for the best variant of each approach (Lee et al. [21], Fabric-VSF [18], and FFN) on both the rectangular cloth and the T-shirt in Supplementary Table S8 in Supplementary Section E. FFN achieves the best performance on generalization to both cloth shapes, by a large margin.

---

### Official Review · Reviewer_a9Nv · 2021-07-24

**Originality:** Good
**Technical Quality:** Fair
**Clarity Of Presentation:** Good
**Impact:** 4

**Recommendation:**

Weak Accept: I recommend accepting the paper, but will not argue for my recommendation if the majority of other reviewers have a different opinion.

**Summary:**

In this work, a goal-conditioned cloth manipulation policy is developed that uses optical flow to improve its performance. Two networks are used: the first network estimates flow between observation and goal images; the second network use flow from the first network to predict pick points and place points. The proposed method was tested both in simulation and on a real robot setup, compared against other model-free and model-based cloth manipulation policies, with promising results shown in the experiments.



**Issues:**

1. In this work, the relationship between observation and the goal and the actions predictions are done separately, how will it affect the performance if two tasks are performed in a network?
2. The ablations: I was not able to find (later found that in appendix) the information on what type of data these experiments were conducted: “one-step”, “multi-step” or “all”; and also how many repetitions were done to achieve the presented results (in table 2). This can be found in Appendix, however, this seems relevant enough to add in the main text shortly.
3. For the real world experiments, how many trials were conducted?
4. Table 1 presents the average value over several experiments conducted, It would be additionally informative to show a standard deviation for each average value.
5. How was alpha taken as 30 for all experiments? For fabrics of different sizes, alpha is to be different, isn't it?
6.	The article is well organized and easy to read, however, in my opinion, the conclusion section could have been elaborated a bit more.
7.	Line 194, I suggest rephrasing the statement, instead “learned in Sec. 3.3” could be “as explained in Sec. 3.3” or “learned as explained in Sec.3.3”.
8.	PickNet – policy network, is composed of two sub-components PickNet1 and PickNet2, this naming convention can be a bit confusing at first glance. This is well explained in the text but without a proper read the PickNet1 and 2 can be considered as some variants of PickNet.
9.	Some information in the paper seems unnecessarily repeated:
a.	Previous methods for goal-conditioning policy have to learn current and goal relationship and reasoning, while the presented method separates these two tasks – this information appears in lines: 126-129, 174-175, and shortly in conclusions (323).
b.	The improvement compared to the previous works – lines: 73, 87, 95, 142. (the context in these usages is sometimes a bit different but it still seems like repeating).


**Reviewer Expertise:**

Very good: Comprehensive knowledge of the area

**Strengths And Weaknesses:**

The main contribution of the paper is the presentation of a flow-based approach for learning goal-conditioned cloth manipulation policies. As mentioned by the authors a common approach for goal-conditioned policy learning requires the network to learn the relationship between observation and the goal and simultaneously to use this relationship to reason over the actions. With their method, these two tasks are separated and the relationship task is performed using the flow network (but this could be better justified, see more comments below).

**Summary Of Recommendation:**

The paper presents an interesting idea for cloth manipulation (cloth folding in this work) by taking optical flow as the input to predict the pick and place points in the manipulation. It is claimed that the proposed method is better predicting the pick and place points compared to the ones having the observation learning and action prediction together. However, this is not well justified as essentially the proposed method can be extended to connecting the Flow network and PickNet together, i.e., learning the relationship between observation and the goal first and then predicting the actions in one network, trained in an end-to-end manner.

---

> ### Author Response · Authors · 2021-08-29
> **Response to Reviewer a9Nv (1/2)**
>
> Thank you for your insightful comments and suggestions. We address each point in detail below.
>
> __Q1__:
> > It is claimed that the proposed method is better predicting the pick and place points compared to the ones having the observation learning and action prediction together. However, this is not well justified as essentially the proposed method can be extended to connecting the Flow network and PickNet together, i.e., learning the relationship between observation and the goal first and then predicting the actions in one network, trained in an end-to-end manner...the relationship between observation and the goal and the actions predictions are done separately, how will it affect the performance if two tasks are performed in a network?
>
> We provide additional experiments to investigate the effect of end-to-end training. First, we try training our FFN architecture end-to-end.  We train this network with the pick losses as well as the flow loss; all losses are allowed to backpropagate through the entire combined network, including through the FlowNet layers. The results on the square towel are presented in the table below and in Supplementary Section F, Table S9 (“JointFFN”).  This variant performs worse than our method (9.28 vs 7.14 on all goals).
>
> We also tried a variant in which we train a single architecture which consists of a FlowNet, a PickNet, and a PlaceNet, trained end-to-end (Table S9, “JointPredictPlace”).  This is similar to the ablation “PredictPlace” in Table 2, except trained end-to-end.  The difference between these methods and FFN is that our method uses flow to determine the place point whereas these variants predict the place point directly. As can be seen, the performance of JointPredictPlace is very poor (22.88 on all goals), significantly worse than our FFN performance (7.14 on all goals).  We also note that the performance of JointPredictPlace is significantly worse than the “PredictPlace” ablation in Table 2, which uses the same architecture but is not trained end-to-end (10.56, from Table 2). Overall, this result, as well as the one in the paragraph above, indicate that end-to-end training leads to significantly worse performance for this task. Our intuition for this is that the flow network should be trained only with the flow loss and that backpropagating gradients from the pick loss into the flow network adds noise and reduces its performance.
>
> Last, we tried variants of the above two architectures with the flow loss removed, to see if we could train these architectures end-to-end with just a single loss at the end, instead of using an intermediate flow loss.  The results, shown in Table S9, are worse for both variants, showing the importance of the intermediate flow loss.
>
> |  Method | One-Step (n=40)  | Multi-Step (n=6)  | All(n=46)  |
> |---|:---:|:---:|:---:|
> |  JointFFN | 7.60±5.62  |  17.53±15.56 |  9.28±9.39 |
> |  JointPredictPlace |  12.90±11.67 | 35.25±19.22  |  22.88±23.24 |
> |  JointFFN, No Flow Loss | 32.41±22.6  | 68.17±50.35 |  37.07±30.34 |
> |  JointPredictPlace, No Flow Loss | 16.31±22.73  | 50.27±31.44  | 24.39±29.77  |
> |  FFN (Ours)  | 4.46±2.62 |  25.04±22.88 |  7.14±11.06 |
>
> __Q2__:
> > The ablations: I was not able to find (later found that in appendix) the information on what type of data these experiments were conducted: “one-step”, “multi-step” or “all”; and also how many repetitions were done to achieve the presented results (in table 2). This can be found in Appendix, however, this seems relevant enough to add in the main text shortly.
>
> We have updated Sections 4.1.2 and the title of Table 2 to clarify that our ablations were averaged over all 46 goals; we also updated Supplementary Section A.2 to clarify that our policy is deterministic and the simulation is near-deterministic, so we only need 1 trial for our simulation experiments (unlike our real world experiments which use 3 trials).
>
> __Q3__:
> > For the real world experiments, how many trials were conducted?
>
> Real world experiments were evaluated for three trials each. This information is in the caption of Supplementary Table S1. We have also now updated Section 4.2.1 in the main paper to clarify this point.
>
> __Q4__:
> > Table 1 presents the average value over several experiments conducted, It would be additionally informative to show a standard deviation for each average value.
>
> We have added standard deviations across goals to the results in Table 1. Note that the standard deviation of our method is quite large for the multi-step goals (and thus also for “all goals”) but relatively small for the one-step goals. The standard deviation was computed across different goals, so this variance reflects the fact that some of the multi-step goals are much harder for our method than others.

---

> > ### Author Response · Authors · 2021-08-29
> > **Response to Reviewer a9Nv (2/2)**
> >
> > __Q5__:
> > > How was alpha taken as 30 for all experiments? For fabrics of different sizes, alpha is to be different, isn't it?
> >
> > The alpha threshold presented at the end of Section 3.5 is provided in pixel coordinates. For a fixed camera and workspace, this threshold can be deprojected from pixels to a fixed metric distance. Alpha represents the distance by which two grippers are likely to collide if they try to grasp points that are a distance of alpha apart. Alpha therefore does not vary with the size of the fabric, unless the camera moves closer or further from the workspace to accommodate fabrics of different sizes (which would change the projection matrix from meters to pixels). For our experiments, the camera height is fixed so we use a fixed alpha for all experiments.
> >
> > __Q6__:
> > > The article is well organized and easy to read, however, in my opinion, the conclusion section could have been elaborated a bit more.
> >
> > We have added discussion on future work to the conclusion. We welcome any additional suggestions on elaborating the conclusion.
> >
> > __Q7__:
> > > Line 194, I suggest rephrasing the statement, instead “learned in Sec. 3.3” could be “as explained in Sec. 3.3” or “learned as explained in Sec.3.3”.
> >
> > We have made this change in the new paper revision.
> >
> > __Q8__:
> > > PickNet – policy network, is composed of two sub-components PickNet1 and PickNet2, this naming convention can be a bit confusing at first glance. This is well explained in the text but without a proper read the PickNet1 and 2 can be considered as some variants of PickNet.
> >
> > We understand that the naming convention for PickNet1 and 2 can be confusing at first glance. The definitions of PickNet1 and 2 are provided in the second paragraph of Section 3.4. In the absence of additional comments or suggestions we plan on keeping the naming convention.
> >
> > __Q9__:
> > > Some information in the paper seems unnecessarily repeated. a. Previous methods for goal-conditioning policy have to learn current and goal relationship and reasoning, while the presented method separates these two tasks – this information appears in lines: 126-129, 174-175, and shortly in conclusions (323). b. The improvement compared to the previous works – lines: 73, 87, 95, 142. (the context in these usages is sometimes a bit different but it still seems like repeating).
> >
> > In the first case (a), we have revised section 3.4 to reduce repetition. For the second case (b), we have also reduced repetition by revising the related work to explain the similarities and differences between different prior approaches as suggested by reviewer RKFg.

---

### Author Response · Authors · 2021-08-31
**Response to Meta-Review and all Reviewers**

(Previously posted as response to meta-reviewer)

We appreciate the feedback from our reviewers to help us improve the manuscript. We are grateful for the positive comments on the clarity of writing, novel application of optical flow to manipulation, and the comprehensiveness of our simulated and real world experiments demonstrating improvement over the state of the art and generalization to unseen cloth shapes.

We have carefully revised and improved the manuscript in response to the questions raised, with the key additional experiments and discussion points listed below:
* Added additional experiments to evaluate separating the flow and pick network vs. training a joint network end-to-end (see response to reviewer a9Nv)
* Addressing the question on whether specifying subgoals is a limitation (see response to reviewer RKFg)
* Addressing the question of how crumpled starting configurations affects performance (see response to reviewer sL8a)
* Updated the writing and organization of the main paper and supplement according to reviewer comments

We have also updated the manuscript with the best results for each method. For FFN and Fabric-VSF, we found that the best performance was obtained by training on data in which the maximum action length was the diagonal of the square cloth (150px vs. 100px in the original manuscript). Lee et al. already used a large action length (156 px) as part of its discrete action space; however, we did see improved performance with Lee et al. when training it with 20k training examples, the same number as FFN. The results of Table 1 and 2, as well as Supplementary Tables S{2,5-10}, have been updated to report the best performance of each method. These results show that FFN outperforms all variants of Fabric-VSF [18] and Lee et al. [21] on average error over all test goals, and that it also outperforms all of the ablations.

The key revisions in the updated manuscript are highlighted in yellow.

We appreciate any additional feedback and will make sure to reflect the comments in the final version.

Best regards,\
Authors

---

### Meta-Review · Area_Chair_xfc1 · 2021-08-13

**Recommendation:** Accept (Poster)
**Confidence:** 5

**Metareview:**

All three reviewers were generally positive about the paper, and the main weaknesses were based on small queries and clarifications, which should be addressed in the rebuttal. Along with these, I would also like the authors to particularly address two issues raised by the reviewers. Firstly, I would like to hear motivation for the need to separate the flow network and the pick network, rather than training a single end-to-end network, as raised by Reviewer a9Nv. Secondly, I would like to hear the authors' option on whether the need to manually specify the sub-goals for a particular task, as mentioned by Reviewer RKFg, is a significant weakness of the work.

-----

After the reviews, authors have provided an updated paper which addresses the reviewers' main issues, including some new experiments. This is a new state-of-the-art method for deformable object manipulation, with good real-world results, and all three reviewers recommend acceptance of the paper.

---

> ### Author Response · Authors · 2021-08-29
> **Response to Area Chair xfc1 and all Reviewers**
>
> We appreciate the feedback from our reviewers to help us improve the manuscript. We are grateful for the positive comments on the clarity of writing, novel application of optical flow to manipulation, comprehensiveness of our simulated and real world experiments demonstrating improvement over the state of the art, and generalizability of our method to unseen cloth shapes.
>
> We have carefully revised and improved the manuscript in response to the questions raised, with the key additional experiments and discussion points listed below:
> * Added additional experiments to evaluate separating the flow and pick network vs. training a joint network end-to-end (see response to reviewer a9Nv)
> * Addressing the question on whether specifying subgoals is a limitation (see response to reviewer RKFg)
> * Addressing the question of how crumpled starting configurations affects performance (see response to reviewer sL8a)
> * Updated the writing and organization of the main paper and supplement according to reviewer comments
>
> We have also updated the manuscript with the best results for each method. For FFN and Fabric-VSF, we found that the best performance was obtained by training on data in which the maximum action length was the diagonal of the square cloth (150px vs. 100px in the original manuscript). Lee et al. already used a large action length (156 px) as part of its discrete action space; however, we did see improved performance with Lee et al. when training it with 20k training examples, the same number as FFN. The results of Table 1 and 2, as well as Supplementary Tables S{2,5-10}, have been updated to report the best performance of each method. These results show that FFN outperforms all variants of Fabric-VSF [18] and Lee et al. [21] on average error over all test goals, and that it also outperforms all of the ablations.
>
> The key revisions in the updated manuscript are highlighted in yellow.
>
> We also appreciate any additional feedback and comments to the end of the discussion phase and we will make sure to reflect the comments in the final version.
>
> Best regards, \
> Authors

---

### Decision · Program_Chairs · 2021-09-13

**Decision:**

Accept (Poster)

**Comment:**

All three reviewers were generally positive about the paper, and the main weaknesses were based on small queries and clarifications, which should be addressed in the rebuttal. Along with these, I would also like the authors to particularly address two issues raised by the reviewers. Firstly, I would like to hear motivation for the need to separate the flow network and the pick network, rather than training a single end-to-end network, as raised by Reviewer a9Nv. Secondly, I would like to hear the authors' option on whether the need to manually specify the sub-goals for a particular task, as mentioned by Reviewer RKFg, is a significant weakness of the work.

-----

After the reviews, authors have provided an updated paper which addresses the reviewers' main issues, including some new experiments. This is a new state-of-the-art method for deformable object manipulation, with good real-world results, and all three reviewers recommend acceptance of the paper.